# GenSE: Generative Speech Enhancement via Language Models using Hierarchical Modeling

**Jixun Yao[1], Hexin Liu[2], Chen Chen[2], Yuchen Hu[2], EngSiong Chng[2], Lei, Xie[1]***
1. Northwestern Polytechnical University 2. Nanyang Technological University

## Abstract

Semantic information refers to the meaning conveyed through words, phrases, and contextual relationships within a given linguistic structure. Humans can leverage semantic information, such as familiar linguistic patterns and contextual cues, to reconstruct incomplete or masked speech signals in noisy environments. However, existing speech enhancement (SE) approaches often overlook the rich semantic information embedded in speech, which is crucial for improving intelligibility, speaker consistency, and overall quality of enhanced speech signals. To enrich the SE model with semantic information, we employ language models as an efficient semantic learner and propose a comprehensive framework tailored for language model-based speech enhancement, called *GenSE*. Specifically, we approach SE as a conditional language modeling task rather than a continuous signal regression problem defined in existing works. This is achieved by tokenizing speech signals into semantic tokens using a pre-trained self-supervised model and into acoustic tokens using a custom-designed single-quantizer neural codec model. To improve the stability of language model predictions, we propose a hierarchical modeling method that decouples the generation of clean semantic tokens and clean acoustic tokens into two distinct stages. Moreover, we introduce a token chain prompting mechanism during the acoustic token generation stage to ensure timbre consistency throughout the speech enhancement process. Experimental results on benchmark datasets demonstrate that our proposed approach outperforms state-of-the-art SE systems in terms of speech quality and generalization capability. Codes and demos are publicly available at `https://yaoxunji.github.io/gen-se`.

## 1 Introduction

Speech enhancement (SE) aims to improve the intelligibility and quality of speech, particularly in situations where degradation is caused by various noises. Most existing SE approaches (Richter et al., 2023; Lemercier et al., 2023; Tai et al., 2024; Yang et al., 2024b) focus on learning a deterministic mapping from noisy to clean speech or modeling the distribution of clean speech. However, these approaches often ignore the rich semantic information embedded in speech, which can be crucial for improving intelligibility, speaker consistency, and overall signal quality. By neglecting semantic information, conventional approaches may experience performance degradation in complex noise conditions or when applied to unseen domains (Kato & Baese-Berk, 2024; Wang et al., 2024c). Integrating semantic information into the SE process can better understand speech attributes, leading to more robust enhancement even in challenging acoustic environments.

Language models (LMs) have demonstrated their superiority in learning and representing semantic information across various natural language processing (NLP) tasks (Lehmann et al., 2023; Jin et al., 2023). Employing LMs in the enhancement process in a generative manner can harness their powerful semantic modeling capabilities, enabling better speech reconstruction beyond simple noise removal. LM-based speech generation approaches first tokenize speech into discrete tokens before training models in a next-token prediction manner. There are two commonly used speech tokens: semantic and acoustic tokens. Acoustic tokens function at a fine granularity, capturing detailed infor-

---
*Corresponding authors

mation about the audio waveform and enabling high-quality reconstruction (Borsos et al., 2023a). In contrast, semantic tokens primarily encode higher-level information, such as phonetics, syntax, and semantics, which is essential for capturing language structure. In the conventional speech-to-speech generation pipeline, a continuous speech signal is first tokenized into semantic tokens, followed by LMs generating acoustic tokens based on these semantic tokens or other instructive information. Finally, acoustic tokens are reconstructed into the speech signal. Semantic tokens are crucial in this process, as they carry essential information about various speech attributes. In the SE task, the input speech is contaminated by noise, which complicates the modeling of both acoustic and semantic aspects. Noise distorts the speech signal, making it difficult to model semantic tokens accurately, which in turn affects the generation of acoustic tokens. Consequently, directly applying generative paradigms designed for clean speech to SE tasks can lead to predictable performance degradation.

On the other hand, acoustic tokens offer the advantage of a reconstruction paradigm, which plays a crucial role in bridging continuous speech and token-based language models. Current acoustic codec models (Défossez et al., 2022; Zeghidour et al., 2021; Zhang et al., 2023) employ a series of quantizers organized in a residual manner to tokenize speech signals. While these models have demonstrated impressive performance, the multiple quantizers result in long acoustic token sequences, significantly increasing the computational expenses and difficulty for LMs. Meanwhile, when multiple quantizers are employed, downstream LMs require additional design efforts to handle the increased complexity of predicting tokens extracted from multiple quantizers. These efforts may include incorporating both autoregressive and non-autoregressive structures (Wang et al., 2023a), parallel generation (Borsos et al., 2023b), or delay prediction (Copet et al., 2024).

In this study, we propose a novel generative framework, GenSE, that leverages LM and discrete speech tokens for speech enhancement. We design a single quantizer codec model with a reorganization process to reduce the number of tokens the LM needs to predict, resulting in a simpler and more efficient prediction process. The reorganization process selects the most frequently used tokens and combines them into a new quantization space, resulting in a larger codebook size without significantly sacrificing performance. Additionally, we introduce a hierarchical modeling method that separates the denoising and generation stages. Specifically, a noise-to-semantic (N2S) module transforms the noisy speech signal into clean semantic tokens, while a semantic-to-speech (S2S) module generates clean acoustic tokens. By decoupling denoising from generation, the N2S module focuses solely on transforming noisy semantic tokens into clean ones, ensuring that noise does not affect the subsequent generation process. This reduces the prediction difficulty for the LM, improving stability and performance in generating high-quality clean speech. We also explore various prompt types to leverage knowledge from language models in our approach, and we propose a token chain prompting mechanism to concatenate noisy and clean semantic tokens along with noisy acoustic tokens to form a comprehensive prompt for the S2S module. This prompting mechanism helps capture more speaker characteristics from the noisy speech, ensuring timbre consistency. The main contributions of this study can be summarized as follows:

- We present GenSE, a novel generative framework designed for LM-based speech enhancement. It employs a series of decoder-only LMs to predict discrete acoustic tokens based on semantic information, significantly improving both enhancement quality and generalization performance, even in challenging noise conditions.

- We propose a simple neural codec model, SimCodec, which incorporates a reorganization process and contains only a single quantizer to reduce the number of acoustic tokens in the temporal dimension. SimCodec achieves remarkable reconstruction quality at a lower bit rate, optimizing both efficiency and performance for downstream LM prediction.

- We introduce a hierarchical modeling method to decouple the denoising process from the generation process and improve the stability of LM prediction. A token chain prompting mechanism is employed to ensure timbre consistency throughout the enhancement process.

## 2 RELATED WORK

### 2.1 GENERATIVE SPEECH ENHANCEMENT

Generative speech enhancement learns the distribution of clean speech as a prior for enhancement, rather than directly mapping noisy speech to clean speech. Most of generative models have been

applied in SE, including generative adversarial networks (GANs) (Fu et al., 2019; Liu et al., 2021), variational autoencoders (VAEs) (Fang et al., 2021; Bie et al., 2022), flow-based models (Nugraha et al., 2020), and diffusion probabilistic models (Lemercier et al., 2023; Tai et al., 2024; Yang et al., 2024b; Scheibler et al., 2024). A series of studies (Le et al., 2024; Yang et al., 2023b; Wang et al., 2024a) have leveraged large-scale datasets to develop audio generation models capable of handling multiple tasks, such as speech synthesis (Zhang et al., 2024), voice conversion (Yao et al., 2024b;a), and speech recognition (Liu et al., 2024b). While these models demonstrate versatility, their performance may be constrained compared to systems specifically optimized and designed for a particular task. Meanwhile, with the growing popularity of discrete representations extracted from neural audio codecs, some works have begun to use discrete tokens for speech enhancement. Genhancer (Yang et al., 2024a) argues that discrete codec tokens offer an efficient latent domain, which can replace the conventional time or time-frequency domain of signals, allowing for more complex modeling of speech. Another work, called SELM (Wang et al., 2024c), is a pioneer in exploring using LMs for speech enhancement. It also employs discrete semantic tokens as the speech representation and incorporates LMs to generate clean semantic tokens conditioned on noisy input. However, these approaches still fall short in terms of speech quality and similarity compared to clean speech.

## 2.2 SPEECH TOKENIZATION

There are two widely used speech tokens in speech language models: semantic tokens and acoustic tokens. Semantic tokens are typically derived from representations produced by self-supervised speech models like HuBERT (Hsu et al., 2021) or WavLM (Chen et al., 2022), and they primarily capture the phonetic and semantic content of speech signals. While semantic tokens are initially designed as training targets for self-supervised models, recent efforts have used them to directly represent high-level semantic information (Borsos et al., 2023a). Acoustic tokens, on the other hand, are extracted from neural audio codec models, which tokenize high-rate audio signals into a finite set of tokens (Défossez et al., 2022). AudioLM (Borsos et al., 2023a) pioneered the use of LMs for audio generation by employing both semantic and acoustic tokens and building several LMs across different stages. VALL-E (Wang et al., 2023a) further extended the AudioLM framework and applied it to text-to-speech (TTS), achieving impressive zero-shot voice cloning. Inspired by VALL-E, several speech-to-speech generation tasks, such as speech translation (Dong et al., 2023) and voice conversion (Wang et al., 2023b), have replaced phonemes with semantic tokens and introduced powerful LMs to achieve remarkable results.

## 2.3 NEURAL AUDIO CODEC

Previous neural audio codec models have been widely used in audio communication to compress audio into discrete representations. Recently, these codec models have gained interest in speech generation tasks, bridging continuous speech and token-based language models. A typical codec model consists of an encoder, a quantization module, and a decoder, with the quantization module playing a crucial role in tokenizing continuous speech features into discrete tokens. Several works explore different architectures to pursue better reconstruction quality (Wu et al., 2023; Kumar et al., 2024; Ai et al., 2024), others enhance the compression rate (Yang et al., 2023a; Ji et al., 2024a;b) or optimize codec space (Ren et al., 2024; Ju et al., 2024; Zhang et al., 2023; Du et al., 2024; Liu et al., 2024a). Most approaches employ residual vector quantization (RVQ), which typically uses eight quantizers, each with a 1024-codebook size, to provide sufficient discrete space for quantization. However, the numerous acoustic tokens generated by these models create significant complexity and challenges in the LM prediction. Although some works (Li et al., 2024; Ji et al., 2024b) explore the use of a single quantizer, their reconstruction quality often falls short or relies on an additional reference encoder to disentangle global representations, which are unsuitable for speech enhancement.

## 3 GENSE

### 3.1 OVERALL ARCHITECTURE

In this section, we introduce GenSE, a cutting-edge system designed to enhance various types of degraded signals, providing better speech quality, speaker similarity, and generalization ability. As

shown in Figure 1, GenSE is composed of three key components: 1) SimCodec, a neural speech codec that extracts acoustic tokens from speech through a quantizer and reconstructs the speech signal from these tokens via a decoder; 2) A pre-trained self-supervised model, XLSR (Conneau et al., 2020), used for extracting high-level semantic information from speech; 3) Two decoder-only LMs that generate clean speech signals conditioned on noisy input. GenSE employs a hierarchical modeling framework with a two-stage process: a N2S transformation front-end, which converts noisy speech into clean semantic tokens, and an S2S generation back-end, which synthesizes clean speech using both semantic tokens and noisy acoustic tokens while preserving the speaker's personalized characteristics. We introduce the SimCodec in Section 3.2 and the hierarchical modeling framework in Section 3.3.

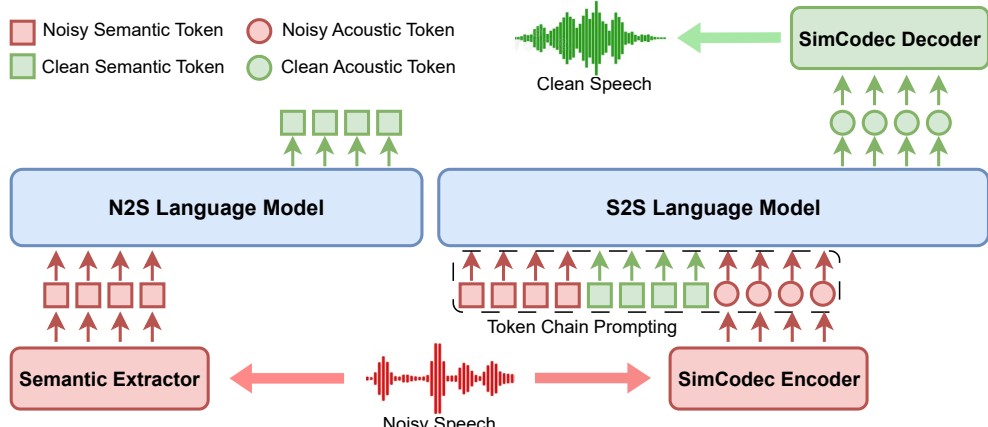

Figure 1: The hierarchical modeling framework of language model in GenSE.

## 3.2 REORGANIZATION PROCESS FOR SIMCODEC

### 3.2.1 SIMCODEC OVERVIEW

We propose an efficient neural codec model, SimCodec, to compress speech signals into discrete tokens by a single quantizer, thereby reducing the number of tokens the LM needs to predict. Sim-Codec is based on the VQVAE architecture (Van Den Oord et al., 2017) and follows the same pattern as EnCodec (Défossez et al., 2022). Specifically, SimCodec consists of three modules: 1) a fully convolutional encoder that compresses the input speech signal into a frame-level latent representation; 2) a quantizer that discretizes the frame-level latent features into discrete tokens using the codebook; and 3) a decoder that mirrors the encoder's structure to reconstruct the speech waveform from discrete tokens.

The quantized discrete token in SimCodec acts as the acoustic token, which serves as the generation target for the S2S module. Unlike most existing codec models that employ multiple quantizers, SimCodec utilizes a single quantizer with a larger codebook size of 8192. While training a model with such a large codebook size often results in low codebook usage and slow convergence, we address these challenges through a specially designed two-stage training process with a codebook reorganization process. This approach improves both codebook usage and reconstruction quality, enabling SimCodec to achieve high performance with fewer tokens.

### 3.2.2 EXPLORATION OF CODEBOOK SIZE AND QUANTIZE SPACE

SimCodec is designed as an acoustic token extractor for speech language models, with a focus on achieving higher reconstruction quality and audio fidelity using fewer tokens. In neural codec models, the quality is primarily determined by the bit rate (Défossez et al., 2022), which is influenced by factors such as the codebook size, token count per second, and the number of quantizers (Mentzer et al., 2023b). Initially, we aim to simplify the model by reducing the number of quantizers from the conventional eight to a single quantizer, while keeping other factors constant. However, this approach resulted in suboptimal performance, with the training process being difficult to converge.

We then consider expanding the codebook size to accommodate the vast space of discrete speech, while still using only a single quantizer. This means that when reducing the number of quantizers from eight to one, we try to expand the codebook size by eight times, even though it doesn't result in the same bit rate. As a result, we progressively increase the codebook size from $2^8$ to $2^{13}$ and evaluate the codebook usage of quantized embeddings and the reconstruction quality.

As shown in Figure 2, we use the Perceptual Evaluation of Speech Quality (PESQ) metric to evaluate the reconstruction quality of Sim-Codec. We observe a clear trend: as the codebook size increases, so does the reconstruction quality. This aligns with expectations from a compression perspective, where a larger codebook size provides more bits to represent information, typically resulting in better reconstruction performance. Specifically, we find that a conventional codec codebook size of $2^{10}$ may not fully utilize the potential of the speech space. When we increase the codebook size from $2^{10}$ to $2^{11}$, the PESQ score improves, indicating better reconstruction quality. However,

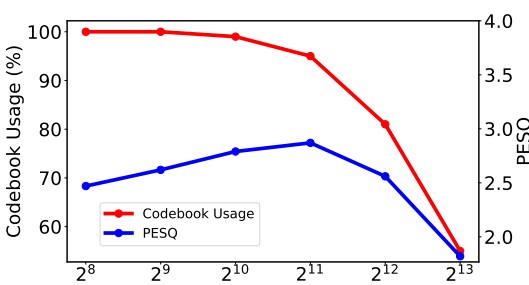

Figure 2: The relationship between reconstruction quality and codebook usage across different codebook sizes.

further increasing the codebook size to $2^{13}$ results in a significant drop in codebook usage from nearly 100% to around 50%. This means that half of the quantized embeddings are left unused in the quantizer, which leads to a degradation in reconstruction quality. This diminishing return at larger codebook sizes in a single quantizer led us to explore a more refined solution.

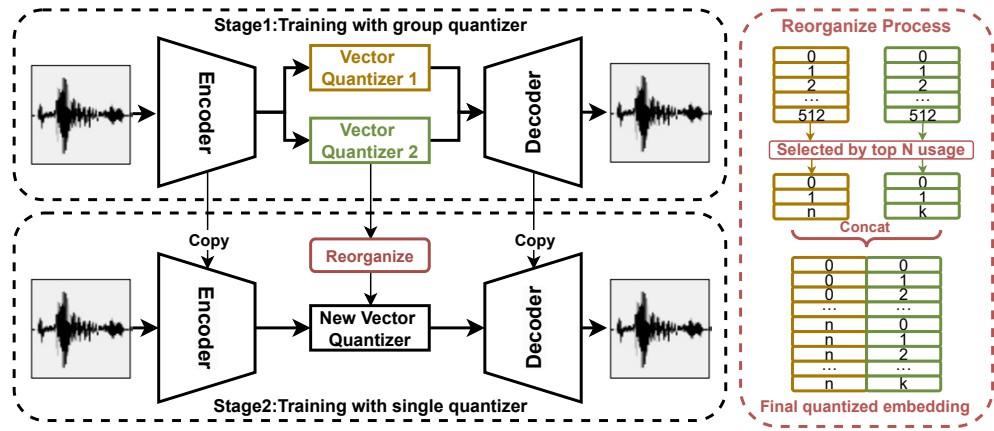

Figure 3: The detailed architecture and training process of SimCodec, with the reorganization process of the group quantizer is highlighted in the red dashed block.

### 3.2.3 CODEBOOK REORGANIZATION

To improve the codebook usage and reconstruction quality with a larger codebook size, we design a two-stage training process for SimCodec, as illustrated in Figure 3. In the first stage, we employ two quantizers with smaller codebook sizes, organized in a group quantization scheme rather than the traditional residual quantization method (Défossez et al., 2022). In conventional residual quantization, the most important information is concentrated in the first quantizer, leading to an unequal distribution of informativeness across subsequent quantizers. This imbalance makes it difficult to merge the two quantizers equitably to create a new quantization space.

After the first stage, we sort the quantized embeddings in each quantizer by codebook usage and select the top $N$ and $K$ quantized embeddings from each quantizer, respectively. We then pairwise concatenate these selected quantized embeddings and reorganize them into a new codebook with a codebook size of $N * K$. In the second stage, we initialize the encoder and decoder parameters to fit the new codebook dimension, while copying the parameters from the first stage for continued

training. At this point, the codec model is nearly fully trained, except for the projection layer adapted to the new codebook. As a result, the training in this stage converges quickly due to the quantized space remains almost identical to the first stage. Additionally, since the quantization process is tied to the dimensional space rather than the temporal domain, the concatenated embeddings will maintain a high codebook usage within a similar quantized space. SimCodec is trained in an end-to-end GAN-based manner, optimizing a reconstruction loss over frequency domains and a perceptual loss using discriminators at different resolutions, details can be found in Appendix A.2.

### 3.3 HIERARCHICAL MODELING FRAMEWORK

To improve the stability of LM prediction, we propose a hierarchical modeling framework with two main components: a N2S transformation front-end, which transforms noisy speech into clean semantic tokens, and an S2S back-end, which synthesizes clean speech while preserving the personalized characteristics of the original speaker. By explicitly decoupling the denoising process from the generation process, noise affecting semantic token extraction is addressed early on, preventing it from influencing speech generation. Figure 1 provides an illustrative overview of GenSE with the hierarchical modeling framework.

#### 3.3.1 NOISY-TO-SEMANTIC TRANSFORMATION

The N2S module first employs a powerful pre-trained self-supervised model to extract semantic tokens from the speech signal. For this purpose, we employ XLSR[1], which is a large-scale multi-lingual pre-trained model for speech representation. XLSR first encodes the speech signal using a stack of convolutional and transformer layers to produce continuous representations at every 20-ms frame and then applies k-means clustering to discretize these representations into a set of cluster indices $S = \{s_1, s_2, ..., s_T\}$. $T$ is the number of frames and $s_T \in [K]$, where $K$ is the number of cluster centroids and we set $K = 1024$.

By compressing speech signals into discrete semantic tokens through the self-supervised model, semantic irrelevant information from continuous speech representations is eliminated, enabling more effective training in an NLP paradigm. Meanwhile, the self-supervised model is also noise-robust to some extent (Chen et al., 2022). The N2S module leverages a language model to learn denoising transformations through a next-token prediction approach. The training objective is as follows:

$$p(x) = \prod_{i=1}^{n} p\left(\hat{s}_n \mid \hat{s}_1, \ldots, \hat{s}_{n-1}, \bar{s}_{1,\ldots,n}\right), \tag{1}$$

where $\hat{s}$ and $\bar{s}$ represent clean semantic tokens and noisy semantic tokens, respectively. $n$ is denote as the frame length, while $\bar{s}$ is concatenated with the $\hat{s}$ and trained in an autoregressive manner.

#### 3.3.2 SEMANTIC-TO-SPEECH GENERATION

The S2S module is responsible for generating clean acoustic tokens and reconstructing clean speech signals using the SimCodec decoder. Instead of directly converting clean semantic tokens into acoustic tokens, we introduce a token chain prompting mechanism in the generation process. This mechanism leverages both the noisy semantic token and the predicted clean semantic token as intermediate prompts, which are concatenated with the noisy acoustic token to generate the clean acoustic token. The acoustic token in the token chain prompting mechanism provides characteristics of the original speaker, helping to ensure consistency of speaker timbre in the enhancement process.

Benefit from the SimCodec, the extracted acoustic token is a single sequence in the temporal dimension, which becomes straightforward to directly connect semantic and acoustic tokens and compute the joint probabilities conditioned on the prompt token:

$$p(x) = \prod_{i=1}^{n} p\left(\hat{a}_n \mid \hat{a}_1, \ldots, \hat{a}_{n-1}, \bar{s}_{1,\ldots,n}, \hat{s}_{1,\ldots,n}, \bar{a}_{1,\ldots,n}\right), \tag{2}$$

where $\hat{a}$ and $\bar{a}$ represent clean acoustic tokens and noisy acoustic tokens, respectively. Once the S2S module generates the clean acoustic tokens, the decoder component of SimCodec reconstructs

---

[1]`https://huggingface.co/facebook/wav2vec2-xls-r-300m`

them back into the enhanced speech signals. The hierarchical modeling method allows each stage to specialize in specific tasks—denoising in the first stage and speech generation in the second stage. As a result, the prediction difficulty for the LM is reduced, leading to improved stability and performance in denoising the degraded speech and generating high-quality speech.

Table 1: Comparison results between our proposed GenSE and various baseline systems. "Noisy" refers to the original degraded signals and higher scores indicate better performance.

| | Without Reverb | | | | | With Reverb | | | | |
| | DNSMOS ↑ | | | SECS ↑ | VQ ↑ | DNSMOS ↑ | | | SECS ↑ | VQ ↑ |
| | SIG | BAK | OVL | | | SIG | BAK | OVL | | |
| Noisy | 3.39 | 2.62 | 2.48 | - | 0.566 | 1.76 | 1.50 | 1.39 | - | 0.527 |
| FullSubNet | $3.05_{+0.0\%}$ | $3.51_{+0.0\%}$ | $2.93_{+0.0\%}$ | 0.62 | 0.623 | $2.74_{+0.0\%}$ | $3.01_{+0.0\%}$ | $2.27_{+0.0\%}$ | 0.63 | 0.615 |
| Inter-Subnet | $3.17_{+3.9\%}$ | $3.15_{-10.3\%}$ | $2.98_{+1.7\%}$ | 0.62 | 0.616 | $2.85_{+4.0\%}$ | $3.09_{+2.7\%}$ | $2.36_{+4.0\%}$ | 0.61 | 0.594 |
| CDiffuSE | $3.37_{+10.5\%}$ | $3.52_{+0.3\%}$ | $2.84_{-3.1\%}$ | 0.58 | 0.624 | $2.82_{+2.9\%}$ | $3.31_{+10.0\%}$ | $2.42_{+6.6\%}$ | 0.57 | 0.611 |
| SGMSE | $3.47_{+13.8\%}$ | $3.41_{-2.8\%}$ | $3.11_{+6.1\%}$ | 0.58 | 0.651 | $3.07_{+12.0\%}$ | $3.02_{+0.3\%}$ | $2.49_{+9.7\%}$ | 0.56 | 0.632 |
| StoRM | $3.54_{+16.0\%}$ | $3.69_{+5.1\%}$ | $3.15_{+7.5\%}$ | 0.61 | 0.687 | $3.02_{+10.2\%}$ | $3.18_{+5.6\%}$ | $2.56_{+12.8\%}$ | 0.62 | 0.646 |
| SELM | $3.47_{+13.8\%}$ | $3.81_{+8.5\%}$ | $3.21_{+9.3\%}$ | 0.62 | 0.673 | $3.04_{+10.9\%}$ | $3.51_{+16.6\%}$ | $2.67_{+17.6\%}$ | 0.61 | 0.659 |
| DOSE | $3.58_{+17.3\%}$ | $3.98_{+13.4\%}$ | $3.31_{+13.0\%}$ | 0.61 | 0.692 | $3.23_{+17.9\%}$ | $\mathbf{3.79}_{+25.9\%}$ | $3.13_{+37.9\%}$ | 0.63 | 0.668 |
| GenSE | $\mathbf{3.65}_{+19.7\%}$ | $\mathbf{4.18}_{+19.1\%}$ | $\mathbf{3.43}_{+17.1\%}$ | **0.67** | **0.717** | $\mathbf{3.49}_{+27.4\%}$ | $3.73_{+23.9\%}$ | $\mathbf{3.19}_{+40.5\%}$ | **0.65** | **0.671** |

## 4 EXPERIMENTS AND RESULTS

### 4.1 EXPERIMENTAL SETUP

**Dataset:** Following previous works (Wang et al., 2024c; Tai et al., 2024), the clean speech data consists of subsets from LibriLight (Kahn et al., 2020), LibriTTS (Zen et al., 2019), VoiceBank (Veaux et al., 2013), and the deep noise suppression (DNS) challenge datasets (Reddy et al., 2021). The noise datasets used are WHAM! (Wichern et al., 2019) and DEMAND (Thiemann et al., 2013). Room impulse responses (RIRs) from openSLR26 and openSLR28 (Ko et al., 2017) are randomly selected to simulate reverberation. All training data are generated on the fly, with an 80% probability of adding noise at a signal-to-noise ratio (SNR) ranging from -5 dB to 20 dB, and a 50% probability of convolving the speech with RIRs. We use the testset from publicly available DNS Challenge (Reddy et al., 2021) to compare GenSE with existing state-of-the-art baseline systems. Following Tai et al. (2024), we use the CHiME-4 dataset (Du et al., 2016) as an additional test set to evaluate the generalization ability of the models. All samples are at a 16kHz sample rate.

**Baseline systems:** We compare our GenSE with recent remarkable and state-of-the-art speech enhancement models containing: FullSubNet (Hao et al., 2021), Inter-Subnet (Chen et al., 2023), CDiffuSE (Lu et al., 2022), SGMSE (Welker et al., 2022), StoRM (Lemercier et al., 2023), SELM (Wang et al., 2024c), and DOSE (Tai et al., 2024). Model configuration and details of baseline systems can be found in Appendix A.3 and Appendix A.4, respectively.

**Evaluation metrics:** Both objective and subjective metrics are used to evaluate the performance between our proposed GenSE and baseline systems.

- For objective metrics, we employ DNSMOS, speaker embedding cosine similarity (SECS), VQScore (Fu et al., 2024) and word error rate (WER) as the evaluation metrics. Specifically: 1) We use the official DNSMOS (Reddy et al., 2022) model from the DNS challenge[2] and VQscore toolkits[3] to evaluate the quality of the enhanced signals. 2) We use the WavLM-TDCNN speaker verification model[4] to measure speaker similarity between enhanced and clean signals. 3) We use a pre-trained ASR model[5] to transcribe the generated speech and compare with ground-truth transcription. Meanwhile, we employ widely used metrics such as PESQ, short-time objective intelligibility (STOI), and Mel-Ceptral Distortion (MCD) for codec evaluation. Additionally, we introduce UTMOS (Saeki et al., 2022),

---

[2]https://github.com/microsoft/DNS-Challenge/tree/master/DNSMOS
[3]https://github.com/JasonSWFu/VQscore
[4]https://github.com/microsoft/UniSpeech/tree/main/downstreams/speaker_verification
[5]https://huggingface.co/facebook/s2t-small-librispeech-asr

an automatic Mean Opinion Score (MOS) prediction system[6], to evaluate the reconstructed quality of codec from a perceptual perspective.

- For subjective metrics, we use the naturalness mean opinion score (NMOS) and the similarity mean opinion score (SMOS) to evaluate the naturalness and speaker similarity of the enhanced signals, respectively. Participants are invited to listen to the enhanced signals and provide their subjective perception scores on a 5-point scale: '5' for excellent, '4' for good, '3' for fair, '2' for poor, and '1' for bad.

## 4.2 EXPERIMENTAL RESULTS ON SPEECH ENHANCEMENT

**Objective Results.** We compare our proposed GenSE with various baseline systems, and the objective metric results are summarized in Table 1. We observe the following: 1) The signals enhanced by GenSE show a significant improvement over the noisy input, achieving an OVL score of 3.43 without reverb and 3.19 with reverb. This indicates that GenSE effectively enhances degraded speech to a much higher quality. 2) GenSE outperforms baseline systems by a substantial margin in evaluation metrics, confirming its superiority in speech enhancement tasks. 3) GenSE achieves an SECS score of 0.67 without reverb and 0.65 with reverb, outperforming all baseline systems. This highlights its ability to preserve speaker identity while enhancing speech quality. These results demonstrate the superiority of GenSE in addressing various forms of degraded speech signals.

Generative models typically aim to fit the distribution of the training samples, which makes them more robust against domain shifts in the input data. We investigate this property of our proposed GenSE and compare the generalization abilities of GenSE to other baseline systems using the CHiME-4 dataset (Du et al., 2016). The CHiME-4 dataset includes real-recorded noises from four distinct real-world environments: street, pedestrian areas, cafeteria, and bus. These environments significantly differ from the training data domain, providing a comprehensive test for evaluating the generalization performance of the models.

Table 2: Comparison results of enhancement performance on the CHiME-4 dataset.

| | DNSMOS $\uparrow$ | | | SECS $\uparrow$ | VQ $\uparrow$ | WER $\downarrow$ |
|---|---|---|---|---|---|---|
| | SIG | BAK | OVL | | | |
| Noisy | 1.71 | 1.45 | 1.36 | - | 0.502 | 37.9 |
| FullSubNet | $2.42_{+0.0\%}$ | $2.71_{+0.0\%}$ | $2.01_{+0.0\%}$ | 0.54 | 0.632 | $34.1_{-10.0\%}$ |
| Inter-Subnet | $2.45_{+1.2\%}$ | $2.69_{-1.8\%}$ | $2.08_{+3.5\%}$ | 0.53 | 0.627 | $29.6_{-21.9\%}$ |
| CDiffuSE | $2.61_{+7.8\%}$ | $2.72_{+0.3\%}$ | $2.38_{+18.4\%}$ | 0.56 | 0.622 | $40.7_{+7.3\%}$ |
| SGMSE | $2.77_{+14.5\%}$ | $2.73_{+0.4\%}$ | $2.41_{+19.9\%}$ | 0.57 | 0.639 | $37.2_{-1.8\%}$ |
| StoRM | $3.21_{+32.6\%}$ | $3.08_{+13.7\%}$ | $2.57_{+27.9\%}$ | 0.55 | 0.645 | $35.1_{-5.8\%}$ |
| SELM | $3.09_{+27.7\%}$ | $3.26_{+20.3\%}$ | $2.62_{+30.3\%}$ | 0.58 | 0.641 | $29.4_{-22.4\%}$ |
| DOSE | $3.17_{+30.9\%}$ | $2.88_{+6.3\%}$ | $2.61_{+29.9\%}$ | 0.61 | **0.654** | $31.2_{-17.7\%}$ |
| GenSE | $\mathbf{3.22}_{+33.1\%}$ | $\mathbf{3.47}_{+28.0\%}$ | $\mathbf{2.89}_{+43.8\%}$ | **0.62** | 0.650 | $\mathbf{28.4}_{-25.1\%}$ |

As shown in Table 2, when there is a domain shift from the DNS test set to the CHiME-4 test set, deterministic-based approaches like FullSubNet and Inter-Subnet experience a significant drop in performance compared to generative-based approaches. The difference in noise characteristics between the training data and the CHiME-4 test set leads to a drastic decline in DNSMOS scores of the enhanced signal, in some cases nearly reaching the levels of the noisy signals. Our proposed GenSE achieves the best performance in both quality and speaker similarity compared with baseline systems, demonstrating more excellent generalization capability against domain shifts in noise characteristics. Additionally, GenSE achieves the lowest WER results, highlighting its ability to enhance speech intelligibility and preserve linguistic information, even in challenging noise environments. While the metrics on the CHiME-4 test set are lower than those on the DNS test set, GenSE shows less degradation compared to other baseline systems, proving its robustness to variations in noise characteristics.

**Subjective Results.** We visualize the subjective evaluation results of speech naturalness and speaker similarity using violin plots, as shown in Figure 4. Participants rated GenSE significantly higher

---

[6]https://github.com/tarepan/SpeechMOS

than all baseline systems in both NMOS and SMOS. The median values for speech naturalness and speaker similarity of signal enhanced by GenSE surpass those of the baseline systems. Additionally, the overall distribution of subjective scores for GenSE outperforms the baselines, further highlighting its superiority in naturalness, quality, and speaker similarity of the enhanced speech. These subjective findings further corroborate the objective results presented in Table 1 and Table 2.

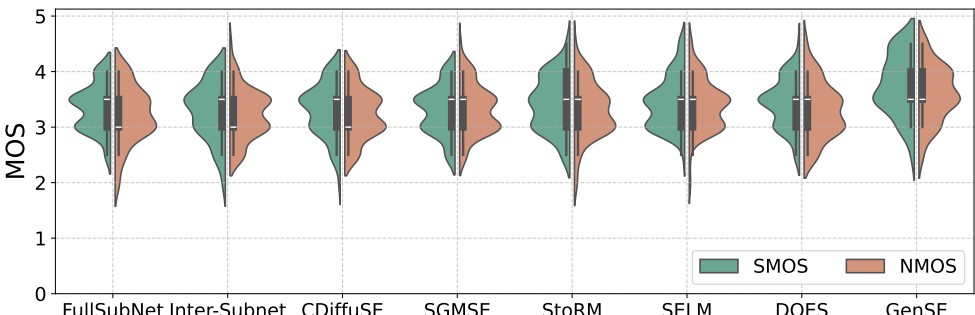

Figure 4: Violin plots for speech naturalness and speaker similarity, comparing the signals enhanced by baseline systems and GenSE. The thin inner line of each violin plot represents the full range of MOS values, while the thick inner line marks the interquartile range (between the first and third quartiles). The white point in each plot indicates the median value.

## 4.3 EXPERIMENTAL RESULTS ON SIMCODEC

In this subsection, we compare the proposed SimCodec in terms of reconstruction quality with strong baselines such as EnCodec (Défossez et al., 2022), HiFi-Codec (Yang et al., 2023a), SpeechTokenizer (Zhang et al., 2023), Vocos (Siuzdak, 2023), and Descript-Audio-Codec (DAC) (Kumar et al., 2024). Table 3 presents the comparison results across different bit rates. Our proposed SimCodec significantly outperforms DAC under similar bandwidth settings (1.73 in PESQ, 0.17 in STOI, 1.09 in MCD, and 1.31 in UTMOS, respectively) and achieves comparable performance with other baselines that operate at higher bit rates. Additionally, SimCodec delivers high-quality reconstruction while using the fewest tokens, significantly reducing the complexity and difficulty of downstream tasks compared to other codec models. We also compare the performance of GenSE using different codec models for speech enhancement, with results provided in Appendix A.5. Moreover, we investigate the performance of replacing our proposed reorganization strategy with different quantization strategies, results are shown in Appendix A.6. These experiments further confirm the necessity and effectiveness of SimCodec within the entire speech enhancement framework.

Table 3: Comparison results of reconstruction quality between our proposed SimCodec and baseline codec models. "Nq" represents the number of the quantizer in the codec model and "token/s" represents the token count per second after quantization.

| Model | Bandwidth ↓ | Nq ↓ | token/s ↓ | PESQ ↑ | STOI ↑ | MCD ↓ | UTMOS ↑ |
|---|---|---|---|---|---|---|---|
| DAC | 9.0 kbps | 9 | 900 | 3.77 | 0.971 | 2.67 | 4.05 |
| SpeechTokenizer | 6.0 kbps | 8 | 600 | 3.25 | 0.957 | 3.06 | 3.57 |
| Encodec | 6.0 kbps | 8 | 600 | 2.92 | 0.946 | 3.51 | 3.41 |
| Vocos | 6.0 kbps | 8 | 600 | 3.37 | 0.961 | 3.22 | 3.48 |
| DAC | 4.0 kbps | 4 | 400 | 2.61 | 0.917 | 4.03 | 3.09 |
| HiFi-Codec | 4.0 kbps | 4 | 400 | 3.17 | 0.959 | 3.66 | 3.48 |
| SpeechTokenizer | 3.0 kbps | 4 | 300 | 2.71 | 0.922 | 4.07 | 3.16 |
| Encodec | 3.0 kbps | 4 | 300 | 1.93 | 0.884 | 4.78 | 2.30 |
| Vocos | 1.5 kbps | 2 | 150 | 1.59 | 0.812 | 4.74 | 2.55 |
| DAC | 1.0 kbps | 1 | 100 | 1.32 | 0.784 | 4.91 | 2.06 |
| SimCodec | 0.65 kbps | 1 | 50 | 2.45 | 0.903 | 3.99 | 3.04 |
| SimCodec | 1.3 kbps | 1 | 100 | 3.05 | 0.954 | 3.82 | 3.37 |

## 4.4 ABLATION AND ANALYZE

We conduct an ablation study to evaluate the performance of removing the token chain prompting mechanism and hierarchical modeling from our GenSE framework, highlighting the contribution of each component to the system's overall performance. Specifically: 1) We only use the clean semantic token to generate the clean acoustic token in the S2S module rather than concatenating the clean semantic token with the noisy semantic token and noisy acoustic token as intermediate prompts. 2) We further remove the hierarchical modeling, employing a single LM with the same model parameters as the combined LMs in N2S and S2S modules to directly predict the clean acoustic token from the noisy semantic token.

Table 4: Results of ablation study on GenSE framework, where "w/o" denotes the absence of a specific method or component.

|  | DNSMOS ↑ | | | SECS ↑ | VQ ↑ |
|---|---|---|---|---|---|
|  | SIG | BAK | OVL |  |  |
| GenSE | $3.57_{+0.0\%}$ | $3.96_{+0.0\%}$ | $3.31_{+0.0\%}$ | 0.66 | 0.649 |
| w/o token chain prompting | $3.54_{-0.8\%}$ | $4.01_{+1.3\%}$ | $3.28_{-0.9\%}$ | 0.43 | 0.636 |
| w/o hierarchical modeling | $3.38_{-5.6\%}$ | $3.75_{-9.6\%}$ | $3.17_{-4.4\%}$ | 0.41 | 0.621 |

As shown in Table 4, removing the token chain prompting mechanism leads to significantly lower SECS results compared to the original GenSE. This indicates a degradation in speaker similarity, with the S2S module failing to maintain speaker consistency after enhancement. This highlights the crucial role of the token chain prompting mechanism in capturing more acoustic characteristics of the original speaker. Furthermore, a notable decline in DNSMOS is observed when switching from hierarchical modeling to a single LM. This confirms the effectiveness of separating denoising and generation into different stages. The hierarchical modeling method not only improves the stability of the generation process but also allows the system to focus more effectively on both semantic and acoustic aspects, resulting in superior performance when enhancing degraded signals. A case comparison of the hierarchical modeling ablation is presented in Appendix A.7.

## 5 CONCLUSION

In this study, we propose GenSE, an LM-based generative SE system with two key components: 1) A novel neural speech codec, SimCodec, with a reorganizing process to improve the codebook usage under a larger codebook size using a single quantizer. The proposed codec delivers high-quality reconstruction with fewer tokens, significantly reducing the complexity and difficulty of downstream tasks. 2) A hierarchical modeling framework designed to enhance degraded signals through a two-stage generative process. In our framework, a N2S front-end transforms noisy speech into clean semantic tokens, and an S2S back-end synthesizes clean speech from the generated clean semantic tokens. We also design a token chain prompting mechanism within the hierarchical modeling framework to maintain speaker consistency. Experimental results demonstrate that GenSE outperforms state-of-the-art speech enhancement systems in both subjective and objective evaluations, demonstrating its effectiveness in speech enhancement. Ablation studies and codec comparison results further confirm the effectiveness of each component in our proposed GenSE framework. Limitations and future works are discussed in Appendix A.8.

### ACKNOWLEDGMENTS

This work was supported by Tencent and Tencent-NTU Joint Research Laboratory (CENTURY), Nanyang Technological University, Singapore.

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

## A    APPENDIX

In the supplemental material:

- A.1. We discuss several common questions for the design of GenSE.
- A.2. We provide training objectives of SimCodec.
- A.3. We provide the model configurations of SimCodec and LM.
- A.4. We describe the details of baseline systems employed in our experiments.
- A.5. We ablate different codec models in our framework and compare the performance on speech enhancement.
- A.6. We compare the performance of using different quantization strategies in the Sim-Codec.
- A.7. We present a visual case of the enhanced sample.
- A.8. We discuss the limitations and future works.

### A.1    ADDITIONAL DISCUSSIONS ON THE DESIGN OF GENSE FRAMEWORK

*Question 1*: *Can GenSE employ different SSL models to extract semantic tokens?*

GenSE is a general framework that can be compatible with various SSL models. We conduct additional experiments to evaluate the enhancement performance when we replace the XLSR in GenSE with the widely used WavLM (Chen et al., 2022), as shown in Table 5. We can find that GenSE achieves similar results when using these two different SSL models.

On the other hand, we chose XLSR as the semantic extractor due to the key advantage of its multilingual speech representation capabilities. While other SSL models also offer robustness to noise and high-quality semantic representations, XLSR stands out because it is pre-trained on a large-scale, multilingual dataset, enabling it to learn robust speech features across different languages. This multilingual generalization ability is highly valuable for SE tasks, which frequently involve diverse languages and varying acoustic environments.

Table 5: Comparison results of employing XLSR and WavLM as semantic extractors in GenSE.

| | DNSMOS ↑ | | | SECS ↑ | VQ ↑ |
|---|---|---|---|---|---|
| | SIG | BAK | OVL | | |
| GenSE with XLSR | 3.57 | 3.96 | 3.31 | 0.66 | 0.694 |
| GenSE with WavLM | 3.48 | 4.05 | 3.25 | 0.65 | 0.712 |

***Question 2***: *Why is the conventional PESQ metric not involved in the evaluation?*

Several speech enhancement papers (Kumar et al., 2020; Liu et al., 2021; Maiti & Mandel, 2020; Li & Yamagishi, 2020) have demonstrated that PESQ may not accurately reflect the true quality of speech, particularly for speech generated by generative models. The key issue lies in the fact that PESQ is an intrusive metric, meaning it evaluates quality by directly comparing the enhanced speech to its clean counterpart. However, generative models aim to model the distribution of real data, and as a result, the generated speech might sound realistic despite having a more considerable distance from the clean version. Therefore, intrusive metrics such as PESQ and STOI may not provide a precise evaluation of signal quality for generative models.

Lower PESQ results do not necessarily indicate that the generated speech has lower quality, as noted by studies such as (Reddy et al., 2022) and (Hsu et al., 2022). To address this limitation, we also conducted an ABX test to assess the perceptual quality of the enhanced speech compared to clean speech, as shown in Figure 5. This evaluation provides a more subjective measure of quality, capturing how close the generated speech sounds to clean speech.

***Question 3***: *How does the performance of GenSE compare with clean samples from a subjective perceived aspect?*

We conduct an ABX test to compare the performance between GenSE and baseline systems, aiming to evaluate human perception and measure how distinguishable the enhanced speech is from clean samples. As shown in Figure 5, GenSE is significantly closer to clean speech samples compared to the baseline systems. Participants in the ABX test consistently preferred the speech enhanced by GenSE over the other baseline systems, indicating that GenSE effectively preserves key characteristics of the original speech, such as naturalness, intelligibility, and speaker consistency. This suggests that the subjective perceptual quality of enhanced signals from GenSE is much higher and closer to the clean samples, demonstrating its effectiveness in handling noisy signals.

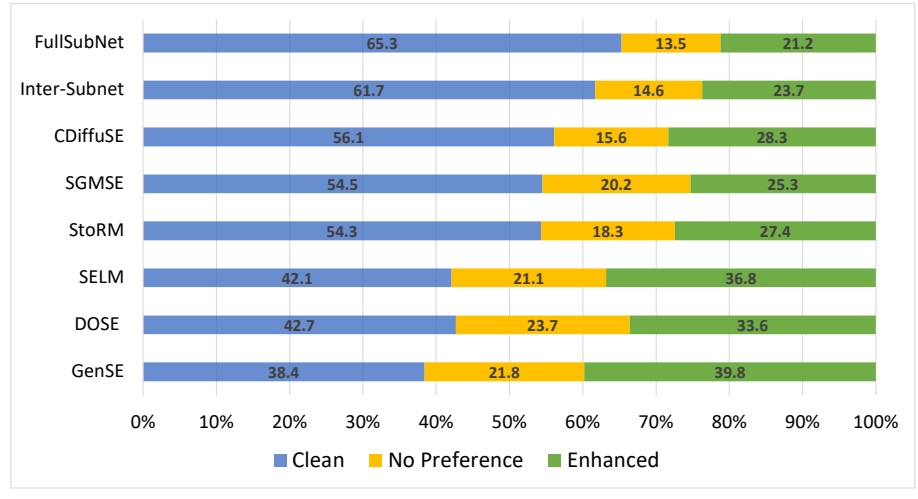

Figure 5: ABX results between GenSE and baseline systems. "No Preference" means that participants perceive the enhanced signal to be as realistic as the clean signal.

***Question 4***: *Can the N2S module in a non autoregressive manner?*

We believe that the generation performance of the N2S module will benefit from autoregressive modeling due to the sequential nature of semantic token generation. The input noisy speech is processed in a step-by-step manner to ensure that the semantic representation remains consistent and aligned with the temporal structure of the speech signal, making autoregressive modeling necessary. To further investigate this, we conducted experiments comparing the performance of the N2S module using autoregressive and non-autoregressive modeling.

Table 6: Comparison results of employing auto-regressive (AR) modeling and non-auto-regressive (NAR) modeling in N2S module.

| | DNSMOS ↑ | | | SECS ↑ | VQ ↑ | Latency ↓ |
|---|---|---|---|---|---|---|
| | SIG | BAK | OVL | | | |
| N2S with AR | 3.57 | 3.96 | 3.31 | 0.66 | 0.694 | - |
| N2S with NAR | 3.35 | 3.79 | 3.03 | 0.63 | 0.662 | -45.7% |

As shown in Table 6, non-autoregressive methods achieve significantly faster inference speeds due to their inherent nature of parallel processing. However, all metrics are degraded compared with autoregressive modeling methods. Non-autoregressive models often struggle with tasks that require capturing complex dependencies or maintaining sequence consistency, such as transforming noisy tokens into semantically meaningful clean tokens. In this case, the autoregressive nature of the N2S module ensures a more precise and stable transformation of noisy semantic tokens into clean ones, which is crucial for the downstream S2S module's generation process.

## A.2 Training objectives of SimCodec

Suppose $X$ is the input speech signal, the total training objectives contain conventional reconstruction loss, a commitment loss for the quantizer, and several discriminator losses with the corresponding feature matching loss. The reconstruction loss is a linear combination between the L1 and L2 losses over the mel-spectrogram using several frequency scales:

$$\mathcal{L}_{\text{rec}} = \sum_{k=0}^{K} (\|\operatorname{STFT}_k(X) - \operatorname{STFT}_k(\hat{X})\|_1 + \|\operatorname{STFT}_k(X) - \operatorname{STFT}_k(\hat{X})\|_2), \quad (3)$$

where $\hat{X}$ is the reconstructed signal and K represents the hop size index. Meanwhile, we can calculate the commitment loss based on the following formula:

$$\mathcal{L}_{\text{com}} = \|z - q(z)\|_2, \quad (4)$$

where $z$ and $q$ represent the hidden output of the encoder and quantizer, respectively. The adversarial losses for the generator $G$ and the discriminator $D$ are defined as follows:

$$\mathcal{L}_{\text{dis}} = \mathbb{E}_{(X,\hat{X})} \left[ (D(X) - 1)^2 + (D(\hat{X}))^2 \right]$$
$$\mathcal{L}_{\text{gen}} = \mathbb{E}_{\hat{X}} \left[ (D(\hat{X}) - 1)^2 \right]. \quad (5)$$

Furthermore, we introduce feature matching loss to measure the difference in intermediate features of the discriminator between a ground truth sample and a generated sample. This loss serves as an additional objective to train the generator. Specifically, every intermediate feature layer of the discriminator is extracted, and the L1 distance between corresponding features from the ground truth sample and the conditionally generated sample is computed in each feature space:

$$\mathcal{L}_{\text{fm}} = \mathbb{E}_{(X,\hat{X})} \left[ \sum_{i=1}^{T} \frac{1}{N_i} \left\| D^i(X) - D^i(\hat{X}) \right\|_1 \right], \quad (6)$$

where $T$ denotes the number of layers in the discriminator; $D_i$ and $N_i$ denote the features and number of features in the i-th layer of the discriminator, respectively. The final objectives are the weighted sum:

$$\mathcal{L}_{\text{total}} = \lambda_1 L_{\text{rec}} + \lambda_2 L_{\text{com}} + L_{\text{gen}} + L_{\text{fm}}, \quad (7)$$

where we set $\lambda_1 = 45$ and $\lambda_2 = 0.1$.

We conduct additional experiments to compare our SimCodec with a contemporaneous work, Wav-Tokenizer (Ji et al., 2024b), which also utilizes a single quantizer. For a fair comparison, we reproduced WavTokenizer using its official code and trained on the same dataset, i.e. LibriSpeech. As shown in Table, our proposed SimCodec (0.65 kbps) outperforms WavTokenizer (0.5 kbps) by a large margin with only 10 additional tokens per second and achieves similar performance compared to WavTokenizer (0.9 kbps). We believe these results can further demonstrate the effectiveness of our SimCodec. On the other hand, our proposed codec supports tokenizing speech with a larger codebook size (8192) compared to WavTokenizer (4096) while using a single tokenizer.

Table 7: Comparison results between our proposed SimCodec and WavTokenizer.

| Model | Bandwidth | Nq | token/s | PESQ | STOI | MCD | UTMOS |
|---|---|---|---|---|---|---|---|
| WavTokenizer | 0.5kbps | 1 | 40 | 1.92 | 0.857 | 4.72 | 2.77 |
| WavTokenizer | 0.9kbps | 1 | 75 | 2.58 | 0.911 | 4.14 | 3.15 |
| SimCodec | 0.65 kbps | 1 | 50 | 2.45 | 0.903 | 3.99 | 3.04 |
| SimCodec | 1.3 kbps | 1 | 100 | 3.05 | 0.954 | 3.82 | 3.37 |

We also visualize the performance between SimCodec and other codec models, as shown in Figure 6.

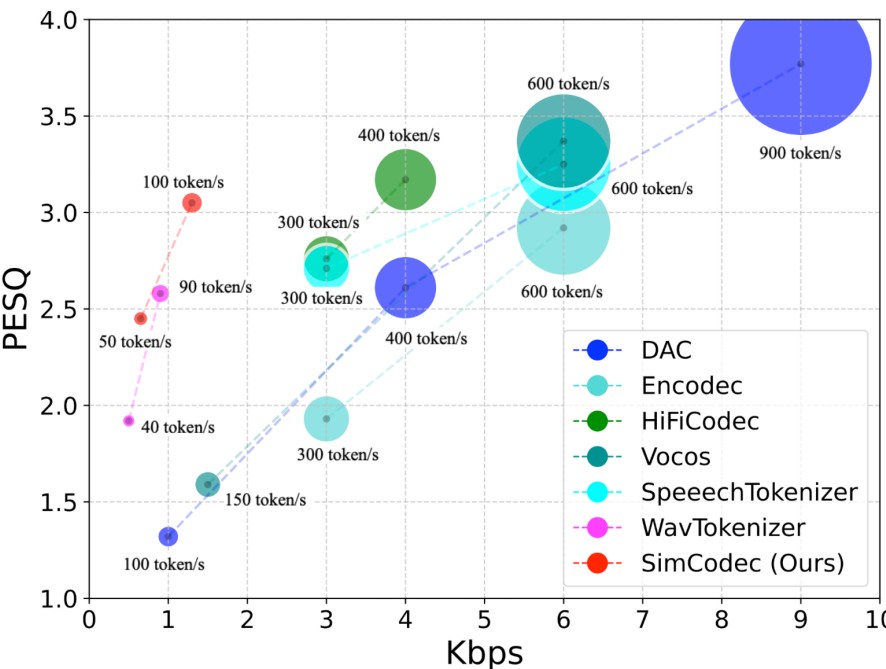

Figure 6: Comparison between SimCodec and state-of-the-art acoustic codec model. The vertical axis PESQ metric measures the reconstructed quality, and the horizontal axis Kbps represents audio compression levels. The size of the circles represents the token numbers per second.

### A.3 MODEL CONFIGURATION

We employ LibriSpeech (Panayotov et al., 2015) for training the SimCodec model, which contains 960 hours of 16kHz speech data. The training is conducted using 2 A100 GPUs with a batch size of 128. The SimCodec model is trained for 50k steps in the first stage and 10k steps in the second stage. We employ the AdamW optimizer with a learning rate of 1e-4 to optimize the codec model. For language model training, we use 8 A100 GPUs with a batch size of 256, training for 1 million steps. We employ the AdamW optimizer with a learning rate of 1e-4 and 5k warmup steps, following the inverse square root learning schedule to adjust the learning rate dynamically during training.

## A.4 Baseline Systems

We compare the performance of our proposed approach with several baseline systems, which we describe in more detail below:

- FullSubNet (Hao et al., 2021): A full-band and sub-band fusion model for single-channel real-time speech enhancement, in which the input full-band and sub-band noisy spectral feature, and output full-band and sub-band speech target.

- Inter-Subnet (Chen et al., 2023): A single-channel lightweight speech enhancement framework employs the subband interaction as a new way to complement the subband model with the global spectral information such as cross-band dependencies and global spectral patterns.

- CDiffuSE (Lu et al., 2022): A speech enhancement algorithm that incorporates characteristics of the observed noisy speech signal into the diffusion and reverse processes.

- SGMSE (Welker et al., 2022): A diffusion speech enhancement framework that does not start the reverse process from pure Gaussian noise but from a mixture of noisy speech and Gaussian noise. This procedure enables using only 30 diffusion steps to generate high-quality clean speech estimates.

- StoRM (Lemercier et al., 2023): A stochastic regeneration approach where an estimate given by a predictive model is provided as a guide for further diffusion, which uses the predictive model to remove the vocalizing and breathing artifacts while producing very high-quality samples thanks to the diffusion model, even in adverse conditions.

- SELM (Wang et al., 2024c): A novel speech enhancement paradigm that integrates discrete tokens and leverages language models, comprises three stages: encoding, modeling, and decoding.

- DOSE (Tai et al., 2024): A model-agnostic diffusion-based speech enhancement method that employs two efficient condition-augmentation techniques.

## A.5 Ablation for Different codec model in SE

To evaluate the necessity of SimCodec in our proposed framework, we replace SimCodec with the current DAC model. We compare the performance of GenSE when employing different codec models with varying numbers of quantizers. For models with more than one quantizer, we explore two prediction strategies: 1) autoregressive and non-autoregressive hybrid prediction as used in (Wang et al., 2023a), and 2) parallel prediction as employed in (Borsos et al., 2023b) for acoustic token generation.

Table 8: Comparison results of GenSE employ different codec models.

| Codec Model | N | Type | DNSMOS ↑ | | | SECS ↑ | VQ ↑ |
|---|---|---|---|---|---|---|---|
| | | | SIG | BAK | OVL | | |
| SimCodec | 1 | Directly | $3.57_{+0.0\%}$ | $3.96_{+0.0\%}$ | $3.31_{+0.0\%}$ | 0.66 | 0.694 |
| DAC | 1 | Directly | $2.39_{-33.1\%}$ | $2.37_{-40.1\%}$ | $2.18_{-34.1\%}$ | 0.35 | 0.547 |
| DAC | 4 | AR+NAR | $3.07_{-14.0\%}$ | $3.46_{-12.6\%}$ | $2.77_{-16.3\%}$ | 0.61 | 0.656 |
| DAC | 4 | Parallel | $3.21_{-10.1\%}$ | $3.57_{-9.8\%}$ | $2.98_{-10.0\%}$ | 0.63 | 0.672 |

As shown in Table 8, we observe the following: 1) Replacing SimCodec with a single-quantizer DAC codec results in a significant decline in both DNSMOS and speaker similarity, highlighting the superiority of SimCodec in achieving high-quality speech enhancement only using a single quantizer. 2) While increasing the number of quantizers improves enhancement performance when using either autoregressive and non-autoregressive hybrid prediction or parallel prediction, the results still do not surpass the performance achieved by employing SimCodec.

Furthermore, we compare the performance between GenSE with current bandwidth and lower bandwidth and we investigate trade-offs between performance gains and computational costs by employing a 50Hz SimCodec (0.65kbps) to replace the current 100Hz (1.3kbps) version for acoustic token extraction. This adjustment reduces the number of acoustic tokens required for prediction by half,

and the number of prefix acoustic tokens needed is also halved, thereby improving computational efficiency.

Table 9: Comparison results between GenSE with different bandwidth.

| Model | SIG | BAK | OVL | SECS | VQ | RTF |
|---|---|---|---|---|---|---|
| GenSE(1.3kbps) | 3.57 | 3.96 | 3.31 | 0.66 | 0.694 | 100% |
| GenSE(0.65kbps) | 3.34 | 3.56 | 3.18 | 0.63 | 0.648 | -24.7% |

As shown in Table 9, we observe a performance degradation in GenSE with lower bandwidth, particularly in DNSMOS metrics (but still outperforms most baseline systems). However, we also computed the real-time factor (RTF) and found that the lower bandwidth version of GenSE achieves over 20% speedup compared to the current version. This improvement is attributed to the significantly reduced number of tokens required for prediction, which enhances prediction efficiency.

Our findings demonstrate that SimCodec offers significant advantages in reducing the number of tokens required for LM prediction while maintaining a competitive level of reconstruction quality compared to higher bit-rate codecs. This reduction in token count simplifies the LM prediction task, contributing to a more efficient and stable speech enhancement process. Moreover, SimCodec's reorganization process and larger codebook size help to further improve performance in terms of both reconstruction quality and compression efficiency.

### A.6 COMPARISON RESULTS FOR USING DIFFERENT QUANTIZATION STRATEGIES IN SIMCODEC.

We investigate the performance of using different quantization strategies and the comparison results are shown in Table. Our proposed reorganization strategy outperforms the clustering vector quantization (CVQ) strategy (Zheng & Vedaldi, 2023) in PESQ, STOI, and MCD metrics, with only a slight degradation in UTMOS. We also observe that finite scalar quantization (FSQ) (Mentzer et al., 2023a) demonstrates lower reconstruction quality. We attribute this to several factors: the smaller latent dimension of the vector in FSQ, the high variance of gradients during training as it approximates hard quantization, and the smooth but less accurate approximation in the early training stages. These challenges are particularly pronounced when employing a single quantizer, potentially limiting FSQ's effectiveness in achieving high-quality reconstruction. An effective group FSQ strategy is employed in (Liao et al., 2024), but it needs several quantizers.

Table 10: Comparison results of using different quantization strategies.

| Model | PESQ | STOI | MCD | UTMOS |
|---|---|---|---|---|
| SimCodec-reorganization | 3.05 | 0.954 | 3.82 | 3.37 |
| SimCodec-CVQ | 2.97 | 0.945 | 3.95 | 3.39 |
| SimCodec-FSQ | 2.51 | 0.913 | 4.53 | 2.94 |

### A.7 VISUALIZE OF THE ENHANCED SPECTROGRAM FROM SEVERELY DEGRADED SAMPLES

We present a case study on a severely noisy signal, as shown in Figure 7. The spectrogram produced by GenSE shows clear harmonic structures and speech formants, demonstrating its ability to effectively suppress noise and retain speech details. In contrast, the variant models without hierarchical modeling exhibit more distortion and residual noise, which indicates inferior noise reduction and signal reconstruction compared to GenSE. This visual comparison further highlights the effectiveness of hierarchical modeling under challenging noise conditions.

### A.8 LIMITATION AND FUTURE WORKS

Despite GenSE has made great progress, it still suffers from the following issues.

**Data coverage.** Although we employ several datasets to train our models, the performance of the GenSE framework could benefit from larger datasets and more model parameters. Expanding the

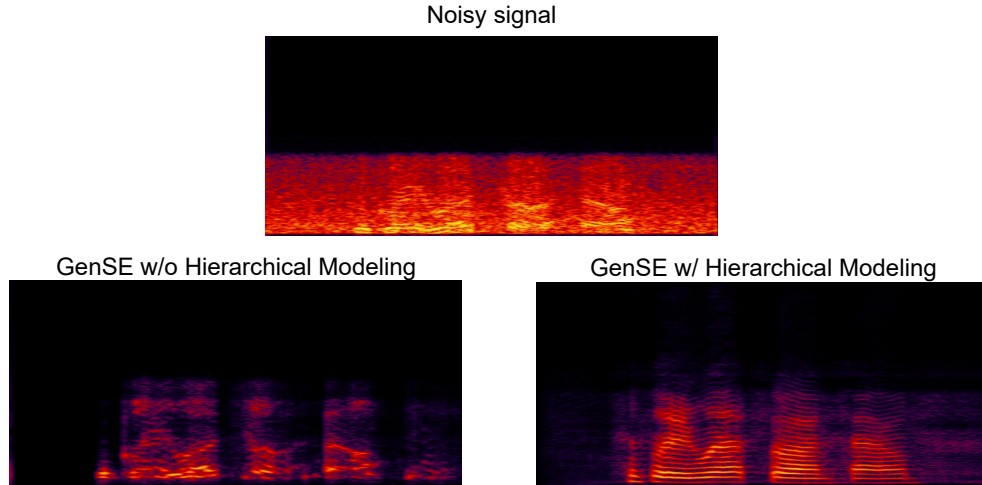

Figure 7: Visualization of the enhanced spectrograms generated by different models for a severely noisy signal.

training data would provide greater coverage of diverse noise types, speaker characteristics, and acoustic environments, thereby further improving the model's generalization to unseen conditions. Additionally, increasing the model's parameters would enable it to capture more complex patterns and nuances in the speech data, further enhancing its ability to produce high-quality, intelligible speech under a wider range of conditions. In the future, we will scale GenSE to more generalized and larger-scale benchmarks to enhance the generation ability.

**Inference efficiency.** Although GenSE outperforms state-of-the-art baselines in both enhancement quality and generalization ability, its inference speed remains a bottleneck due to the autoregressive generation process. This slows down the system, limiting its real-time applicability in practical scenarios. To address this, we believe that the architecture can remain unchanged to support real-time applications by modifying the token prediction pattern during training. Specifically, we propose alternating token prediction in the order of $[s_1, a_1, s_2, a_2, ..., s_n, a_n]$ to replace the current sequential prediction pattern $[s_1, s_2...s_n, a_1, a_2, ..., a_n]$. This approach has been demonstrated effectively in streaming voice conversion (Wang et al., 2024b) and real-time spoken language modeling[7], suggesting its potential to achieve real-time performance within our framework. Furthermore, model quantization and acceleration techniques are necessary for GenSE to be effectively deployed in real-world applications. In future work, we plan to explore efficient strategies to improve inference speed, such as model pruning, or lightweight architectures, enabling faster and more efficient speech enhancement.

---

[7]https://github.com/THUDM/GLM-4-Voice/blob/main/README_en.md

