# OpenReview forum: "GenSE: Generative Speech Enhancement via Language Models using Hierarchical Modeling"
_ICLR.cc/2025/Conference — ICLR 2025 Poster_

### Official Review · Reviewer_fv2J · 2024-11-02

**Soundness:** 2
**Presentation:** 3
**Contribution:** 3
**Rating:** 6
**Confidence:** 5

**Summary:**

This paper proposes a novel approach to speech enhancement (SE) called GenSE, which integrates semantic information into the enhancement process using language models (LMs). Traditional SE methods often ignore the semantic context, focusing solely on mapping noisy to clean speech, which can lead to performance issues in challenging environments. GenSE redefines SE as a conditional language modeling task by leveraging LMs to predict discrete acoustic tokens based on semantic information. It also separates the denoising and generation stages, improving prediction stability and incorporating a token chain prompting mechanism to maintain timbre consistency. The proposed SimCodec Model achieves remarkable reconstruction quality at a lower bit rate. Experimental results show that GenSE outperforms existing SE systems, demonstrating improved intelligibility and robustness in noisy conditions.

**Strengths:**

1. The proposed hierarchical modeling method that separates the denoising and generation stages is effective.
2. The proposed SimCodec reduces the number of tokens in the generation process, which would benefit all speech generation tasks.
3. The experimental results and demo audios are promising.

**Weaknesses:**

**The main issues with this paper lie in the design of SimCodec and the lack of some experimental details:**
*SimCodec*:
1. The issue of low codebook usage with a large codebook size has been identified in the field of computer vision for a long time, and there are already many solutions available [1, 2]. Although this work proposes the codebook reorganization strategy to solve this issue, there are no ablation comparisons between this strategy and baselines like CVQ [2] and FSQ [3]. These comparisons are important for validating the effectiveness of the reorganization strategy proposed in this paper.
2. The codebook reorganization strategy employs two quantizers at the first stage and concat the two quantizers at the second stage. This process is slightly similar to the GRVQ technique of Hifi-Codec [4] and the multichannel quantization of MoVQ [5]. I think the comparative experimental results of these two techniques should be added to Table 3. And the authors should discuss how their approach differs from or improves upon GRVQ and MoVQ.
3. I think Figure 6 looks extremely similar to Figure 1 in WavTokenizer [8], even the colors of the baselines are the same. However, this paper does not compare with WavTokenizer. An explanation why WavTokenizer was not included in the comparison and how their work differs from or builds upon WavTokenizer is required, or 2) Include WavTokenizer as a relevant baseline.

*Some experimental details*:
1. Real-time generation is crucial for speech enhancement models, but the experiments of this paper do not mention the real-time factor (RTF) of the GenSE model. While Table 4 demonstrates that token chain prompting and hierarchical modeling are highly effective, it also does not indicate how much delay these methods introduce.
2. In Section 3.3.2, the prefix token of GenSE at the S2S stage contains noisy acoustic tokens, clean semantic tokens, and noisy semantic tokens, which significantly increase the sequence length in training and inference. This paper lacks a specific analysis of the trade-offs between performance gains and computational costs of the introduced prefix sequence.
3. Mapping from semantic to acoustic using a flow-matching model has proven to be highly effective in many previous studies [6, 7].  The authors could explain why they chose their current approach instead of a flow-matching model for the S2S module, discussing potential advantages and disadvantages. Alternatively, they might consider implementing a flow-matching model as an additional baseline in their experiments to compare its performance with their current method.

**Minor questions that would not influence the scores:**
1. Do you use greedy decoding for decoder LM? Will beam search improve the performance of the model?

**Minor clarity issues**:
1. In Section 3.2.3, Line 264, ``we reinitialize the encoder and decoder parameters to fit the new codebook dimension, while copying the parameters from the first stage``, the use of "reinitialize" in the first half of the sentence introduces clarity issues;
2. In Section 3.3.1, Line 293, ``Meanwhile, the self-supervised model is also noise-robust to some extent.`` Some citations can be added here to demonstrate that this phenomenon actually exists.

**Minor typos**:
1. In Section 1, Line 052, the quotes of ``textless NLP``;
2. In Figure 6, `Our` -> `Ours`.

**Conclusion**:
The SimCodec and hierarchical modeling method proposed in this paper are not particularly novel, as there have been related studies in fields such as Computer Vision and Speech Generation. However, the experimental results are still quite impressive. If the authors could address my concerns, I would increase the score.

[1] Yu, Jiahui, et al. "Vector-quantized image modeling with improved vqgan." arXiv preprint arXiv:2110.04627 (2021).
[2] Zheng, Chuanxia, and Andrea Vedaldi. "Online clustered codebook." Proceedings of the IEEE/CVF International Conference on Computer Vision. 2023.
[3] Mentzer, Fabian, et al. "Finite scalar quantization: Vq-vae made simple." arXiv preprint arXiv:2309.15505 (2023).
[4] Yang, Dongchao, et al. "Hifi-codec: Group-residual vector quantization for high fidelity audio codec." arXiv preprint arXiv:2305.02765 (2023).
[5] Zheng, Chuanxia, et al. "Movq: Modulating quantized vectors for high-fidelity image generation." Advances in Neural Information Processing Systems 35 (2022): 23412-23425.
[6] Du, Zhihao, et al. "Cosyvoice: A scalable multilingual zero-shot text-to-speech synthesizer based on supervised semantic tokens." arXiv preprint arXiv:2407.05407 (2024).
[7] Anastassiou, Philip, et al. "Seed-TTS: A Family of High-Quality Versatile Speech Generation Models." arXiv preprint arXiv:2406.02430 (2024).
[8] Ji, Shengpeng, et al. "Wavtokenizer: an efficient acoustic discrete codec tokenizer for audio language modeling." arXiv preprint arXiv:2408.16532 (2024).

**Questions:**

My questions are included in the weaknesses part.

---

> ### Author Response · Authors · 2024-11-19
> **Response to Reviewer fv2J (1/2)**
>
> We sincerely appreciate you for considering that our work is effective and experimental results are promising. Now we will address your concerns point by point:
>
> **Q1: Although this work proposes the codebook reorganization strategy to solve this issue, there are no ablation comparisons between this strategy and baselines like CVQ [2] and FSQ [3]. These comparisons are important for validating the effectiveness of the reorganization strategy proposed in this paper.**
>
> Thanks for pointing out this point. We investigate the performance of using different quantization strategies and the comparison results are shown as follows:
>
> |Model|PESQ|STOI|MCD|UTMOS|
> |:-:|:-:|:-:|:-:|:-:|
> |SimCodec-reorganization|3.05|0.954|3.82|3.37|
> |SimCodec-CVQ|2.97|0.945|3.95|3.39|
> |SimCodec-FSQ|2.51|0.913|4.53|2.94|
>
> Our proposed reorganization strategy outperforms the CVQ strategy in PESQ, STOI, and MCD metrics, with only a slight degradation in UTMOS. We also observe that FSQ demonstrates lower reconstruction quality. We attribute this to several factors: the smaller latent dimension of the vector in FSQ, the high variance of gradients during training as it approximates hard quantization, and the smooth but less accurate approximation in the early training stages. These challenges are particularly pronounced when employing a single quantizer, potentially limiting FSQ's effectiveness in achieving high-quality reconstruction. A effectiveness group FSQ strategy is employed in [1], but it need several quantizers. We will add these results in the final revision.
>
> [1] Liao, Shijia, et al. "Fish-Speech: Leveraging Large Language Models for Advanced Multilingual Text-to-Speech Synthesis." arXiv preprint arXiv:2411.01156 (2024).
>
> **Q2: The codebook reorganization strategy is slightly similar to the GRVQ technique of Hifi-Codec [4] and the multichannel quantization of MoVQ [5]. I think the comparative experimental results of these two techniques should be added to Table 3. And the authors should discuss how their approach differs from or improves upon GRVQ and MoVQ.**
>
> - Thanks for your comments. The comparison results of HiFi-Codec are already presented in Table 3. The key difference between our proposed codec and HiFi-Codec lies in the distribution of information across quantizers. Unlike HiFi-Codec, which uses a group residual quantization scheme that concentrates the most important information in the first group quantizer, our approach employs a group quantization scheme without residual designed to ensure an equal distribution of informativeness across two quantizers. This strategy avoids the hierarchical dependency of residual quantization and ensures a more balanced representation in the encoded tokens.
> - We believe that MoVQ is an effective quantization approach for image generation. However, there is currently no convincing evidence to suggest that it performs well in speech tokenization. For this reason, we did not include a comparison with MoVQ in Table 3. Our focus remains on evaluating our approach specifically tailored or proven effective for our proposed speech enhancement framework.
>
> **Q3: I think Figure 6 looks extremely similar to Figure 1 in WavTokenizer [8], even the colors of the baselines are the same. However, this paper does not compare with WavTokenizer. An explanation why WavTokenizer was not included in the comparison and how their work differs from or builds upon WavTokenizer is required, or 2) Include WavTokenizer as a relevant baseline.**
>
> We are sorry for the missing reference of WavTokenizer. The evaluation metric in Figure 6 is PESQ, a reference-based distance metric rather than a MOS prediction metric used in WavTokenizer. While WavTokenizer is a solid work, it is also one of the submissions for ICLR 2025, released in August. According to ICLR guidance, we are not required to compare our work with this paper at this stage. However, we are glad to cite this paper and provide a comparative analysis of our codec and WavTokenizer in the Appendix, following your valuable suggestions. The results are as follows:
>
> |Model|Bandwidth|Nq|token/s|PESQ|STOI|MCD|UTMOS|
> |:-:|:-:|:-:|:-:|:-:|:-:|:-:|:-:|
> |WavTokenizer|0.5kbps|1|40|1.92|0.857|4.72|2.77|
> |WavTokenizer|0.9kbps|1|75|2.58|0.911|4.14|3.15|
> |SimCodec|0.65 kbps|1|50|2.45|0.903|3.99|3.04|
> |SimCodec|1.3 kbps|1|100|3.05|0.954|3.82|3.37|
>
> As shown in the table, our proposed SimCodec (0.65 kbps) outperforms WavTokenizer (0.5 kbps) by a large margin with only 10 additional tokens per second and achieves similar performance compared to WavTokenizer (0.9 kbps). We believe these results can further demonstrate the effectiveness of our SimCodec. On the other hand, our proposed codec supports tokenizing speech with a larger codebook size (8192) compared to WavTokenizer (4096) while using a single tokenizer.

---

> ### Author Response · Authors · 2024-11-19
> **Response to Reviewer fv2J (2/2)**
>
> **Q4: Real-time generation is crucial for speech enhancement models, but the experiments of this paper do not mention the real-time factor (RTF) of the GenSE model. While Table 4 demonstrates that token chain prompting and hierarchical modeling are highly effective, it also does not indicate how much delay these methods introduce.**
>
> We acknowledge that the current language model-based approach and hierarchical modeling result in a RTF exceeding 1, which limits its suitability for real-time inference. However, we believe that the architecture can remain unchanged to support real-time applications by modifying the token prediction pattern during training. Specifically, we propose alternating token prediction in the order of $[s_1, a_1, s_2, a_2, ..., s_n, a_n]$ instead of the current sequential prediction pattern $[s_1, s_2...s_n, a_1, a_2, ..., a_n]$. This approach has been demonstrated effectively in streaming voice conversion and real-time spoken language modeling, suggesting its potential to achieve real-time performance within our framework.
>
> **Q5: In Section 3.3.2, the prefix token of GenSE at the S2S stage contains noisy acoustic tokens, clean semantic tokens, and noisy semantic tokens, which significantly increase the sequence length in training and inference. This paper lacks a specific analysis of the trade-offs between performance gains and computational costs of the introduced prefix sequence.**
>
> Thanks for your comments. We have demonstrated the effectiveness and necessity of our proposed token chain prompting through ablation studies, where performance degradation occurs when the prompting tokens are removed. Furthermore, we investigate trade-offs between performance gains and computational costs by employing a 50Hz SimCodec to replace the current 100Hz version for acoustic token extraction. This adjustment reduces the number of acoustic tokens required for prediction by half, and the number of prefix acoustic tokens needed is also halved, thereby improving computational efficiency. The results are as follows:
>
> |Model|SIG|BAK|OVL|SECS|VQ|RTF|
> |:-:|:-:|:-:|:-:|:-:|:-:|:-:|
> |GenSE(100Hz)|3.57|3.96|3.31|0.66|0.694|100%|
> |GenSE(50Hz)|3.34|3.56|3.18|0.63|0.648|-24.7%|
>
> We employ the 100Hz version as the baseline and measure the relative decrease in the real-time factor (RTF) for comparison. Our experiments find that the 50Hz version of GenSE achieves over a 20% speedup compared to the 100Hz version. While there is a performance degradation, it remains within an acceptable margin and still outperforms most baseline systems, demonstrating an effective trade-off between computational efficiency and performance. We will add this analysis in the final revision.
>
> **Q6: Mapping from semantic to acoustic using a flow-matching model has proven to be highly effective in many previous studies [6, 7]. The authors could explain why they chose their current approach instead of a flow-matching model for the S2S module, discussing potential advantages and disadvantages. Alternatively, they might consider implementing a flow-matching model as an additional baseline in their experiments to compare its performance with their current method.**
>
> Thank you for your comments. We acknowledge that employing a flow-matching module and a vocoder often leads to better speech quality in many speech synthesis works. However, speech synthesis typically benefits from complete linguistic content information, which can be partially masked or missing in speech enhancement scenarios. The primary motivation of our work is to leverage semantic information to reconstruct incomplete or masked speech signals in noisy environments. In such cases, autoregressive modeling of semantic tokens excels by capturing both local dependencies (e.g., phonetic features in speech) and global long-term structures (e.g., language syntax and semantic content), which are crucial for enhancing degraded signals. Moreover, employing a flow-matching module generally requires explicit conditioning, which can be challenging to extract from noisy speech waveforms.
>
> **Q7: Do you use greedy decoding for decoder LM? Will beam search improve the performance of the model?**
>
> Thank you for your comments. In our experiments, beam search achieved similar performance to greedy decoding, with some metrics slightly lower than those for greedy decoding.
>
> **Q8: Minor clarity issues and Minor typos**
> - For the minor issues, we will revise our paper following your comments.
>
> The authors thanks again for your efforts and constructive comments for our paper. We hope that the above discussion can address the reviewer's concern and we would be delighted to receive any additional suggestions or comments.

---

> > ### Comment · Reviewer_fv2J · 2024-11-19
> >
> > After reading the authors feedback, I believe they have addressed most of my concerns. The results presented in Q3 highlight SimCodec's performance, which would be a valuable contribution to speech community. Consequently, I have raised my score from 5 to 6.

---

> > > ### Author Response · Authors · 2024-11-19
> > >
> > > Thank you for increasing the score! We sincerely appreciate your constructive suggestions and your recognition of this work.

---

> ### Comment · Reviewer_fv2J · 2024-11-21
> **Question about SimCodec**
>
> I have one more question regarding the results of SimCodec: in the Table of Q3, the UTMOS scores for WavTokenizer show significant degradation compared to the results reported in the WavTokenizer paper. The authors should provide more experimental details and clarify the reasons behind the discrepancies in UTMOS scores.
>
> | Models                        | UTMOS from authors | UTMOS from WavTokenizer's paper | Difference |
> | :------------:                 | :------------: | :------------: | :------------: |
> | WavTokenizer-0.5kps  | 2.77 | 3.6016 | - 0.8316 |
> | WavTokenizer-0.9kps  | 3.15 | 4.0486 | - 0.8986 |

---

> > ### Author Response · Authors · 2024-11-22
> >
> > Thanks for your comments. We train WavTokenizer using the official github repo with the same training data as our SimCodec. We believe two differences exist between our trained WavTokenizer and the one presented in the original paper: 1) **Training Data**: The original WavTokenizer is trained on a larger and more diverse dataset than our reproduced, contributing to its performance advantages. 2) **Quality of Evaluation Samples(more critical)**: The speech quality of the ground truth samples in the evaluation dataset plays a critical role. Higher-quality samples typically lead to better reconstruction results, a phenomenon also observed in WavTokenizer. To clarify this point, we add LibriSpeech and LJSpeech dataset as the evaluation dataset, consistent with WavTokenizer, to re-evaluate and compare performance. The comparison results of UTMOS across different datasets are as follows:
> >
> > |     Model    | Bandwidth |  DNS | LibriSpeech | LJSpeech | Average|
> > |:------------:|:---------:|:----:|:-----------:|:--------:|:----:|
> > | WavTokenizer | 0.5kbps   | 2.77 |     3.04    |   3.89   | 3.20 |
> > | WavTokenizer | 0.9kbps   | 3.15 |     3.32    |   4.07   | 3.51 |
> > | SimCodec     | 0.65 kbps | 3.04 |     3.13    |   3.91   | 3.36 |
> > | SimCodec     | 1.3 kbps  | 3.37 |     3.44    |   4.05   | 3.62 |
> >
> > We will include a comprehensive discussion of WavTokenizer in the Appendix of our revised submission to clarify this point thoroughly. We sincerely thank you once again for your valuable comments and the effort you have dedicated to reviewing our work.

---

> > > ### Comment · Reviewer_fv2J · 2024-11-22
> > >
> > > I appreciate the authors' clarification regarding UTMOS and their efforts in conducting additional experiments.

---

### Official Review · Reviewer_tRzd · 2024-11-04

**Soundness:** 3
**Presentation:** 4
**Contribution:** 3
**Rating:** 6
**Confidence:** 4

**Summary:**

The paper introduces GenSE, a generative speech enhancement (SE) framework that integrates language models (LM) to leverage semantic information for enhancing speech signals. Unlike traditional SE methods that focus on signal mapping, GenSE treats SE as a conditional language modeling task. By tokenizing speech into semantic and acoustic tokens using a novel codec (SimCodec) and employing a hierarchical approach, GenSE aims to maintain speaker consistency and improve speech quality under noisy conditions. Experiments demonstrate GenSE’s significant improvements over state-of-the-art SE systems in both quality and robustness to noise.

**Strengths:**

1. GenSE offers a unique perspective by reframing SE as a language modeling task, using semantic information to enhance robustness. This represents a notable departure from conventional deterministic mapping in SE.
2. The hierarchical modeling method, separating semantic and acoustic token generation, improves both quality and intelligibility of enhanced speech, as evidenced by superior metrics across DNSMOS and SECS.
3. The authors present a detailed breakdown of the methodology and technical architecture, providing clear diagrams and tables that make complex processes accessible.
4. By addressing the limitations of traditional SE approaches in handling complex noise environments, GenSE has the potential to impact real-world applications in noisy and challenging acoustic settings.

**Weaknesses:**

1. The hierarchical design and multiple components in GenSE, while effective, may pose a challenge in real-time applications. Simplifying or optimizing these processes further could improve usability.
2. Although SimCodec effectively reduces token count, further exploration into balancing token complexity and quality in low-bandwidth scenarios could enhance GenSE’s adaptability.
3. The two-stage quantizer reorganization might benefit from more empirical comparisons with other single-quantizer methods such as WavTokenizer, as these details are relatively underexplored.

**Questions:**

1. Could the authors provide more insight into how SimCodec might perform under different network conditions, especially with low latency or limited bandwidth?
2. How does the system handle speaker identity in cases of domain shifts, such as across different languages, accents, and ages, and would an alternative to XLSR affect GenSE’s generalization capability?
3. For practical implementation, are there considerations for reducing the computational overhead of the hierarchical modeling method, perhaps through model pruning or compression techniques?

---

> ### Author Response · Authors · 2024-11-19
> **Response to Reviewer tRzd (1/2)**
>
> We sincerely appreciate you for considering that our work offers a unique perspective and has the potential to impact real-world applications. We respond your comments as follows:
>
> **Q1: The hierarchical design and multiple components in GenSE, while effective, may pose a challenge in real-time applications. Simplifying or optimizing these processes further could improve usability.**
>
> For real-time applications, we believe the current architecture can remain unchanged, with modifications applied to the token prediction pattern during training. Specifically, we propose alternating token prediction in the order of $[s_1, a_1, s_2, a_2, ..., s_n, a_n]$ instead of the current sequential prediction pattern $[s_1, s_2...s_n, a_1, a_2, ..., a_n]$. This adjustment aligns with recent approaches demonstrated in streaming voice conversion [1] and real-time spoken language models [2], enabling our framework to support streaming inference in real-time applications. We are confident that this modification provides the necessary adaptability for real-time performance.
>
> [1] Zhichao Wang, et. al. StreamVoice: Streamable Context-Aware Language Modeling for Real-time Zero-Shot Voice Conversion. ACL 2024, pages 7328–7338.
>
> [2] https://github.com/THUDM/GLM-4-Voice/tree/main
>
> **Q2: Although SimCodec effectively reduces token count, further exploration into balancing token complexity and quality in low-bandwidth scenarios could enhance GenSE’s adaptability.**
>
> We compare the performance between GenSE with current bandwidth and lower bandwidth, as shown in the follows:
>
> |Model|SIG|BAK|OVL|SECS|VQ|RTF|
> |:-:|:-:|:-:|:-:|:-:|:-:|:-:|
> |GenSE(1.3kbps)|3.57|3.96|3.31|0.66|0.694|100%|
> |GenSE(0.65kbps)|3.34|3.56|3.18|0.63|0.648|-24.7%|
>
> We observe a performance degradation in GenSE with lower bandwidth, particularly in DNSMOS metrics (but still outperforms most baseline systems). However, we also computed the real-time factor (RTF) and found that the lower bandwidth version of GenSE achieves over 20% speedup compared to the current version. This improvement is attributed to the significantly reduced number of tokens required for prediction, which enhances processing efficiency. We will add these experiments in the Appendix.
>
> **Q3: The two-stage quantizer reorganization might benefit from more empirical comparisons with other single-quantizer methods such as WavTokenizer, as these details are relatively underexplored.**
>
> We add additional comparison with WavTokenizer in Table 3, the added results are as follows:
>
> |Model|Bandwidth|Nq|token/s|PESQ|STOI|MCD|UTMOS|
> |:-:|:-:|:-:|:-:|:-:|:-:|:-:|:-:|
> |WavTokenizer|0.5kbps|1|40|1.92|0.857|4.72|2.77|
> |WavTokenizer|0.9kbps|1|75|2.58|0.911|4.14|3.15|
> |SimCodec|0.65 kbps|1|50|2.45|0.903|3.99|3.04|
> |SimCodec|1.3 kbps|1|100|3.05|0.954|3.82|3.37|
>
> As shown in the table, our proposed SimCodec (0.65 kbps) outperforms WavTokenizer (0.5 kbps) by a large margin with only 10 additional tokens per second and achieves similar performance compared to WavTokenizer (0.9 kbps). We believe these results can further demonstrate the effectiveness of our SimCodec.

---

> ### Author Response · Authors · 2024-11-19
> **Response to Reviewer tRzd (2/2)**
>
> **Q4: Could the authors provide more insight into how SimCodec might perform under different network conditions, especially with low latency or limited bandwidth?**
>
> SimCodec achieves an bandwidth of 0.65 Kbps at 50Hz token generation, which is significantly lower than most current speech codec models. We also investigated further compression to 25Hz, achieving a bandwidth of 0.325 Kbps. However, this led to a significant degradation in reconstruction performance. Reducing the codebook size from 8192 to 2048 resulted in a modest bandwidth reduction from 0.65 Kbps to 0.55 Kbps, but the compression space was significantly constrained, and reconstruction quality deteriorated notably. Furthermore, the architecture of encoder and decoder in SimCodec is similar to pioneering works like EnCodec, enabling support for streaming inference to meet low-latency requirements in real-time applications.
>
> **Q5: How does the system handle speaker identity in cases of domain shifts, such as across different languages, accents, and ages, and would an alternative to XLSR affect GenSE’s generalization capability?**
>
> - In our framework, speaker identity is implicitly preserved through the hierarchical modeling of semantic and acoustic tokens. While domain shifts, such as variations in languages, accents, or age, can pose challenges, our system leverages the pre-trained self-supervised model and in-context learning capabilities. This design ensures desirable generalization when handling speakers with different genders, ages, and languages. For highly divergent domains, we believe in stronger performance with training data that encompasses a broader range of diversity.
> - As an alternative to XLS-R, we have presented the results of replacing XLS-R with WavLM in Appendix A.1 Q1. The findings indicate that GenSE achieves comparable performance when utilizing either of these self-supervised learning models, demonstrating the flexibility of our framework in adopting different SSL models.
>
> **Q6: For practical implementation, are there considerations for reducing the computational overhead of the hierarchical modeling method, perhaps through model pruning or compression techniques?**
>
> In this work, our primary focus was on establishing the effectiveness of the hierarchical modeling framework in speech enhancement. However, we acknowledge that computational efficiency is a critical consideration for practical implementation. To address this, we believe we can adopt techniques such as model pruning, quantization, and knowledge distillation. These methods have shown promise in reducing model complexity while maintaining performance in similar tasks.
>
> We appreciate your suggestion and will incorporate these aspects into our future research directions to make the model more feasible for deployment.

---

> > ### Comment · Reviewer_tRzd · 2024-11-20
> >
> > Thanks for your explanations. I intend to keep my score unchanged.

---

### Official Review · Reviewer_KVEG · 2024-11-04

**Soundness:** 3
**Presentation:** 3
**Contribution:** 2
**Rating:** 6
**Confidence:** 2

**Summary:**

This paper presents GenSE, a novel generative framework for speech enhancement that leverages language models (LMs) and discrete speech tokens. GenSE employs a single-quantizer neural codec model called SimCodec to extract acoustic tokens from speech, reducing the complexity compared to previous multi-quantizer codecs. It also introduces a hierarchical modeling approach that separates the denoising and generation stages, with a noise-to-semantic (N2S) module transforming noisy speech into clean semantic tokens, and a semantic-to-speech (S2S) module generating clean acoustic tokens.

**Strengths:**

1. The proposed generative framework leverages language models and discrete speech tokens to outperform state-of-the-art speech enhancement systems in terms of speech quality and generalization capability.
2. The paper introduces a hierarchical modeling approach that separates the denoising and generation stages, improving the stability and performance of the LM-based generation process.
3. The paper is clearly written and easy to follow.

**Weaknesses:**

The ablation studies are relatively insufficient. For example, it would be helpful to provide detailed analysis on what information are contained in noisy/clean semantic tokens and noisy/clean acoustic tokens, respectively.

**Questions:**

Could you provide a comparison between SimCodec and Vocos (Siuzdak, 2023) and WavTokenizer(Ji, 2024)?

---

> ### Author Response · Authors · 2024-11-19
> **Response to Reviewer KVEG**
>
> We sincerely appreciate you for considering that our work is clearly written and easy to follow. We respond your comments as follows:
>
> **Q1:The ablation studies are relatively insufficient. For example, it would be helpful to provide detailed analysis on what information are contained in noisy/clean semantic tokens and noisy/clean acoustic tokens, respectively.**
>
> - Thanks for your comments. The only difference between clean tokens and noisy tokens is the presence of non-vocal elements, such as background noise, electrical noise, or music, which are absent in clean tokens.
> - The definition and extraction of the semantic and acoustic token follow the pioneer work [1], details as follows: 1) Acoustic tokens operate at a fine level, capturing detailed audio waveform information, and enabling high-quality reconstruction; 2) Coarse-level semantic tokens primarily encode phonetics, syntax, and semantics-related information. Autoregressive modeling of semantic tokens captures both local dependencies (e.g., phonetic features in speech) and global long-term structures (e.g., language syntax and semantic content). However, semantic tokens result in poor reconstruction quality. We will add the details in the final revision.
>
> [1] Borsos, Zalán, et al. "AudioLM: a language modeling approach to audio generation." IEEE/ACM transactions on audio, speech, and language processing 31 (2023): 2523-2533.
>
> **Q2:Could you provide a comparison between SimCodec and Vocos (Siuzdak, 2023) and WavTokenizer(Ji, 2024)?**
>
> We will add a comparison between SimCodec, Vocos, and WavTokenizer in Table 3. The added results are summarized in the table below:
>
> |Model|Bandwidth|Nq|token/s|PESQ|STOI|MCD|UTMOS|
> |:-:|:-:|:-:|:-:|:-:|:-:|:-:|:-:|
> |Vocos|6.0 kbps|8|600|3.37|0.961|3.22|3.48|
> |Vocos|1.5 kbps|2|150|1.59|0.812|4.74|2.55|
> |WavTokenizer|0.5kbps|1|40|1.92|0.857|4.72|2.77|
> |WavTokenizer|0.9kbps|1|75|2.58|0.911|4.14|3.15|
> |SimCodec|0.65 kbps|1|50|2.45|0.903|3.99|3.04|
> |SimCodec|1.3 kbps|1|100|3.05|0.954|3.82|3.37|
>
> As shown in the table, Vocos (6 kbps) achieves better performance than others due to its use of more quantizers. However, there is a significant degradation in performance with Vocos (1.5 kbps). Meanwhile, SimCodec (0.65 kbps) outperforms WavTokenizer (0.5 kbps) by a large margin with only 10 additional tokens per second and achieves similar performance compared to WavTokenizer (0.9 kbps). We believe these results can further demonstrate the effectiveness of our SimCodec.
>
> Once again, we sincerely thank you for your valuable efforts. We would be delighted to receive any additional suggestions or comments.

---

### Official Review · Reviewer_bQqp · 2024-11-05

**Soundness:** 2
**Presentation:** 3
**Contribution:** 2
**Rating:** 6
**Confidence:** 5

**Summary:**

- This paper introduces a language model-based generative speech enhancement system, termed GenSE.
- The system comprises two primary components: a decoder-only model that enhances noisy tokens into clean tokens, and a neural speech codec, SimCodec, which reconstructs waveforms from the enhanced clean tokens.

**Strengths:**

- The paper is clearly written and easy to follow.
- The proposed approach demonstrates the effectiveness of the decoder-only architecture for conventional signal processing tasks, such as speech enhancement (SE).

**Weaknesses:**

- The proposed approach lacks significant novelty, which is the primary reason for my decision to reject the paper. However, please correct me if I am mistaken, as I am open to revisiting my assessment.

- Concerning speech enhancement (SE) using language models (or the decoder-only architecture), similar approaches have already been introduced in:

[1] Wang, X., Thakker, M., Chen, Z., Kanda, N., Eskimez, S. E., Chen, S., ... & Yoshioka, T. (2024). Speechx: Neural codec language model as a versatile speech transformer. IEEE/ACM Transactions on Audio, Speech, and Language Processing.
[2] Yang, D., Tian, J., Tan, X., Huang, R., Liu, S., Chang, X., ... & Meng, H. (2023). Uniaudio: An audio foundation model toward universal audio generation. arXiv preprint arXiv:2310.00704.

Neither of these references are cited.

- Similarly, with regard to the neural speech codec, an analogous method was proposed in:

[3] Li, H., Xue, L., Guo, H., Zhu, X., Lv, Y., Xie, L., ... & Li, Z. (2024). Single-Codec: Single-Codebook Speech Codec towards High-Performance Speech Generation. arXiv preprint arXiv:2406.07422.

This work is also not referenced. Given these omissions, I judge the paper as lacking sufficient originality for acceptance. I believe all referenced works were available prior to the ICLR submission.

**Questions:**

- The lack of novelty mentioned in the weaknesses section diminishes the overall contribution of this paper. Without a substantially innovative approach, I am inclined to recommend rejection.

---

> ### Author Response · Authors · 2024-11-19
> **Response to Reviewer bQqp**
>
> We sincerely appreciate your recognition that our work is clearly written and easy to follow. We will add the missing references following your suggestions, and we address your concerns in detail:
>
> **Q1: Concerning speech enhancement (SE) using language models (or the decoder-only architecture), similar approaches have already been introduced in UniAudio and SpeechX**
>
> Although the S2S module in our system shares conceptual or architecture similarities with UniAudio and SpeechX, all inspired by pioneering works like VALL-E and AudioLM, its goals and model design differ significantly.  We clarify key differences as follows:
>
> - The key difference is that UniAudio and SpeechX aim to build an audio generation model suited for multiple tasks, using a single language model to directly generate acoustic representations from text or other discrete tokens. In contrast, our approach focuses on leveraging semantic information in speech to enhance degraded signals. We introduce a hierarchical modeling method that decouples the generation of clean semantic tokens and clean acoustic tokens into two distinct stages: noise-to-semantic transformation and semantic-to-speech generation. This hierarchical modeling framework is a significant departure from UniAudio and SpeechX and stands as one of the core contributions of our work. Furthermore, our ablation study demonstrates that the hierarchical modeling method outperforms the use of a single language model in speech enhancement.
> - There are also significant differences in acoustic token prediction between our proposed method and UniAudio and SpeechX. SpeechX follows the pattern used in VALL-E, using an autoregressive approach to predict the first layer of acoustic tokens and then predicting the acoustic tokens of other layers in parallel. UniAudio, on the other hand, employs a multi-scale Transformer to predict multi-layer acoustic tokens. In contrast, our system benefits from the proposed SimCodec, where the acoustic token is a single sequence in the temporal dimension, enabling direct prediction and reducing complexity compared to tokens extracted from multiple quantizers.
> - The performance of UniAudio and SpeechX in the specific task of speech enhancement may be suboptimal, as they only compare with early-stage works like DCCRN (Hu et al., 2020) and SGMSE+ (Richter et al., 2022). In contrast, we demonstrate the superior performance of our proposed system compared to recent state-of-the-art speech enhancement studies.
>
> Therefore, both the motivation and contributions of our work are distinct from those of works like UniAudio or SpeechX, and we believe it fits well with ICLR, a conference that encourages innovation.
>
> **Q2: Similarly, with regard to the neural speech codec, an analogous method was proposed in SingleCodec**
>
> Although SingleCodec is also a single quantizer codec model similar to our proposed SimCodec, there are two significant differences:
> - SingleCodec is a mel codec, where the input to the codec encoder is a mel spectrogram rather than a waveform. While mel representations operate at the frame level and are easier to train, they lose some information, resulting in a lower upper bound for reconstructed quality in SingleCodec. This limitation in reconstructed quality has also been reported in [1]. In contrast, our proposed SimCodec directly compresses the waveform and employs a two-stage training strategy with a quantizer reorganization process to address training convergence issues, achieving better reconstruction quality.
> - SingleCodec requires an additional reference encoder to disentangle time-invariant acoustic information from the discrete token sequence. However, this approach can lead to incomplete information being represented by the discrete tokens. This limitation becomes especially pronounced in noisy signals, where the reference encoder faces challenges in extracting accurate acoustic information necessary for reliable waveform reconstruction. In contrast, our proposed SimCodec directly compresses the waveform into discrete tokens without relying on an auxiliary reference encoder, ensuring a more robust representation of acoustic information even in noisy conditions.
>
> We hope that the above discussion can clarify the reviewer's misunderstandings and address proposed concerns. We would be delighted to receive any additional suggestions or comments.
>
> [1] Ji, Shengpeng, et al. "Wavtokenizer: an efficient acoustic discrete codec tokenizer for audio language modeling." arXiv preprint arXiv:2408.16532 (2024).

---

> > ### Comment · Reviewer_bQqp · 2024-11-19
> >
> > It is not good practice to overlook related research while focusing solely on the strengths of the proposed system, especially when similar ideas are shared. I appreciate the inclusion of detailed comparisons, and I recommend incorporating all these comparisons into the paper to ensure readers are well-informed about the historical context of this work. I have adjusted my score to 6, thank you.

---

> > > ### Author Response · Authors · 2024-11-19
> > >
> > > Thank you very much for increasing the score and for your constructive suggestions! We will include these comparisons in the revised version.

---

### Author Response · Authors · 2024-11-26
**Summary of the rebuttal revision**

We thank all the reviewers for their efforts and constructive suggestions for our paper. We summarize the main revision of the manuscript according to the comments and suggestions of reviewers:
- In Section 1, we include definitions and differences between semantic tokens and acoustic tokens.
- In Section 2.1, we add an introduction of generative audio language models, highlighting their objectives and methodologies, and emphasize the differences between our framework and these models.
- In Section 2.3, we add the discussion of the single quantizer codec model.
- In Section 4.4, we add comparison results of Vocos as additional baseline in Table 3.
- In Appendix 2, we present a detailed performance comparison between our proposed SimCodec and WavTokenizer. While WavTokenizer is a contemporaneous work (a submission to ICLR 2025), we added these results to improve the soundness of our proposed SimCodec following reviewers' valuable suggestions.
- In Appendix 5, we investigate the trade-offs between performance gains and computational costs of GenSE under different bandwidths.
- In Appendix 6, we compare the performance of using different quantization strategies in the SimCodec.
- In Appendix 8, we provide a detailed discussion of potential strategies to enhance the practicability of our framework for real-world applications.

These revisions have addressed the reviewers' concerns while strengthening the paper's contributions and evaluation rigor. We believe these changes have markedly improved the manuscript's quality and clarity. We greatly appreciate the reviewers' great efforts and valuable comments, which have significantly improved the soundness of our manuscript.

---

### Meta-Review · Area_Chair_hQX8 · 2024-12-07

**Metareview:**

The paper is clearly written and easy to follow, with detailed breakdowns and clear diagrams. It introduces a hierarchical modeling method that separates denoising and generation stages, improving stability and performance, and reframes speech enhancement (SE) as a language modeling task. The proposed generative framework outperforms state-of-the-art SE systems in terms of speech quality and generalization, with promising experimental results and demo audios.

**Additional Comments On Reviewer Discussion:**

The reviewers raised concerns 1) the paper lacks detailed analysis on the information contained in noisy/clean semantic and acoustic tokens; 2) The hierarchical design and multiple components may pose challenges for real-time use, requiring further simplification or optimization; 3) The proposed approach lacks significant novelty. These issues have been addressed during the author-reviewer discussion.

---

### Decision · Program_Chairs · 2025-01-22

Accept (Poster)